# Observing polarization patterns in the collective motion of nanomechanical arrays

Juliane Doster [1,7], Tirth Shah [2,3,7], Thomas Fösel[2,3], Philipp Paulitschke[4], Florian Marquardt[2,3] & Eva M. Weig [1,5,6 ✉]

In recent years, nanomechanics has evolved into a mature field, and it has now reached a stage which enables the fabrication and study of ever more elaborate devices. This has led to the emergence of arrays of coupled nanomechanical resonators as a promising field of research serving as model systems to study collective dynamical phenomena such as synchronization or topological transport. From a general point of view, the arrays investigated so far can be effectively treated as scalar fields on a lattice. Moving to a scenario where the vector character of the fields becomes important would unlock a whole host of conceptually interesting additional phenomena, including the physics of polarization patterns in wave fields and their associated topology. Here we introduce a new platform, a two-dimensional array of coupled nanomechanical pillar resonators, whose orthogonal vibration directions encode a mechanical polarization degree of freedom. We demonstrate direct optical imaging of the collective dynamics, enabling us to analyze the emerging polarization patterns, follow their evolution with drive frequency, and identify topological polarization singularities.

[1] University of Konstanz, Department of Physics, Universitätsstr. 10, 78457 Konstanz, Germany. [2] Max Planck Institute for the Science of Light, Staudtstr. 2, 91058 Erlangen, Germany. [3] Friedrich-Alexander University Erlangen-Nürnberg (FAU), Department of Physics, Staudtstr. 7, 91058 Erlangen, Germany. [4] Ludwig-Maximilians-Universität Munich, Department of Physics, Geschwister-Scholl-Platz 1, 80539 München, Germany. [5] Technical University of Munich, Department of Electrical and Computer Engineering, Hans-Piloty-Str. 1, 85748 Garching, Germany. [6] Munich Center for Quantum Science and Technology (MCQST), Schellingstr. 4, 80799 München, Germany. [7] These authors contributed equally: Juliane Doster, Tirth Shah. ✉email: eva.weig@tum.de

When the vectorial character of electromagnetic waves was established in the 19th century, this opened the door to the interpretation of a wealth of important phenomena, launching the field of polarization physics. Surprisingly, the detailed nature of spatially inhomogeneous polarization patterns in wave fields began to be analyzed only much more recently. Careful theoretical studies revealed, among other aspects, features such as the topological robustness of certain polarization singularities[1,2]. Overall, this appreciation of the spatial patterns observed in polarization fields has opened a novel domain of inquiry that continues to draw fresh attention and enables modern applications, e.g. tailoring and understanding polarized emission and scattering patterns in nano-optics[3–6].

In recent years, the field of nanomechanics has had wide-ranging impact from sensing applications[7–9] to fundamental physics[10–12]. However, in the world of nanomechanical resonators, it remains challenging to observe polarization physics, even at the level of a single resonator. In the mechanical domain, "polarization" refers to the excitation of motion along different directions. Observing nontrivial effects requires that these vibrational modes are at least almost degenerate, i.e. a geometry with a high degree of symmetry. This condition has been accomplished in macroscopic mechanical setups, such as pendula, which have actually been used in a different domain of experiments, studying transport in arrays[13,14]. It is significantly more demanding in the nanoscopic regime, even though benefits like fast response, high integration density, and the possible combination with other measurement and actuation modalities (optomechanics, surface acoustic waves, embedded quantum dots) create a strong incentive to enter that domain. For nanomechanical resonators, pioneering experimental works have observed two degenerate orthogonal modes with strong coupling within string resonators or nanowires[8,9,15], as well as nonlinear coupling between two near-degenerate orthogonal modes in a nanowire[16]. Going from one or a few such resonators with polarization degrees of freedom to an entire coupled array would enable accessing the wealth of phenomena in polarization fields that have so far only been studied for electromagnetic waves. In recent years, coupled nanomechanical arrays have emerged as a promising platform for observing collective phenomena and transport [17–25]. However, what has been missing so far is a successful integration of polarization degrees of freedom into an array of coupled resonators.

In view of the goal to observe and study polarization patterns, an important aim (besides large-scale integration and coupling) is the ability to easily visualize the motion, in a spatially resolved way. This rules out stiff resonators such as nanobeams or -strings, which, as a result of their small vibrational amplitudes need to be measured individually by very sensitive optical or electrical means and where imaging could at best be achieved in a slow sequential fashion in a scanning tip approach.

On the other hand, nanopillar resonators[8,9,26–28] offer large flexural motion in two orthogonal directions, and have thus been proposed[29] as a natural candidate for rapid spatially resolved optical whole-array imaging of polarization patterns.

## Results

**Description of the sample and measurement technique.** In this work, we investigate an array of 400 nanomechanical pillar resonators (Fig. 1a, b). Each nanopillar exhibits two almost-degenerate orthogonal fundamental flexural vibration modes. The mean frequency and mean anisotropy of the array amount to $\omega \approx 2\pi \times 1.3$ MHz and $\Delta \approx 2\pi \times 7.5$ kHz, respectively. Coupling between adjacent nanopillars via the strain mediated by the substrate has recently been demonstrated[27], joining a small

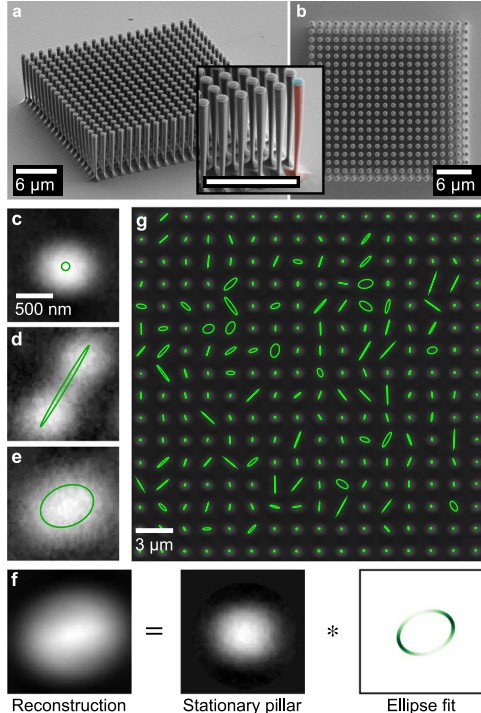

**Fig. 1 Characterization of the sample and spatially resolved imaging of vibrational motion. a, b** Scanning electron micrographs of a 20 × 20 nanopillar array with lattice constant 1.4 μm, pillar diameter $d \approx 300$ nm (measured at the bottom), height $H \approx 6.5$ μm and taper angle $\varphi \approx 1.5°$ in a 60° tilted view and in top view. The inset in **a** shows a zoom of the array corner in the tilted view. False-colors on a single pillar indicate the inverted conical GaAs pillar (red) and the SiO₂ etch mask (blue). **c–e** show top view images of three different pillars (extracted trajectories added in green) differentiating between **c** a pillar at rest and vibrating pillars with **d** linear trajectory and **e** elliptical trajectory. **f** Visualization of the convolution equation. The reconstructed moving pillar image (here, shown for the pillar in **e**) is a convolution (denoted by *) of the stationary (non-driven) pillar image and the time-average of the fitted elliptical trajectory; details can be found in Supplementary Note 2 and Supplementary Figs. 2 and 3. **g** Top view of the 16 × 16 central pillars of the array in **a** driven at $\Omega = 2\pi \times 1.3694$ MHz including trajectories (green). The outermost two rows are omitted here.

number of platforms in which strong coupling between nanomechanical resonators was successfully explored[22–24,30–34]. The coupling strength can be engineered by adjusting the pillar geometry as well as the separation of the pillars. Here, the geometrical parameters of the array are optimized for both large coupling rate $J \approx 2\pi \times 10$ kHz and vibration amplitude $x \approx 0.5$ μm.

The nanopillars are driven at a variable frequency $\Omega/2\pi$ using a piezo actuator. Due to the large vibration amplitudes of the pillar heads even in the linear response regime, the envelope of their trajectories can be captured by optical imaging from the top (for details see Supplementary Note 1 and Supplementary Fig. 1). The optical imaging allows for the simultaneous detection of up to several thousands of nanopillars and their spatial trajectories as a function of frequency, whereas typical measurement techniques for resonator arrays rely on sequential measurements of every single resonator[28] or compromise by giving up spatial resolution[17]. In the resulting micrographs, a pillar at rest appears as a bright circle (Fig. 1c), whereas a vibrating pillar is swept along its trajectory during the imaging process, yielding the envelope of its motion pattern (Fig. 1d, e). We reconstruct the trajectory by demanding that its convolution with the image

of a resting pillar reproduces the observation (Fig. 1f; cf. Supplementary Note 2 and Supplementary Figs. 2 and 3 for details on the algorithm). The extracted trajectories range from linear to elliptical (Fig. 1d, e).

The variety of motional patterns observed in the whole array (Fig. 1g) indicates a certain amount of disorder. Even as a result of minute geometrical variations arising during fabrication, nanoresonators, though nominally identical, typically show a spread in their eigenfrequencies $\delta\omega \approx 2\pi \times 25$ kHz (Supplementary Notes 7 and 8; Supplementary Figs. 6, and 7; Supplementary Table 1). Nonetheless, and despite their narrow linewidth of roughly $\Gamma \approx 2\pi \times 5$ kHz at ambient conditions, a large group of nanopillars vibrates at the same drive frequency. This already suggests that the array exhibits collective motion, which will be demonstrated in more detail later on.

**Polarization physics in a single nanopillar.** We note that elliptical trajectories are observed despite a linear drive, and we will now briefly describe the physics behind that for a single pillar, before moving on to the dynamics of the entire array. When applying an external drive, it will generally excite both linear polarizations with displacements $x$ and $y$, respectively. In general, due to fabricational anisotropies, the underlying eigenmodes will have different resonance frequencies $\omega_{x,y}$. As discussed in the following, this leads to a phase lag in the response to the drive, which can create elliptical motion.

It is convenient to employ complex notation,

$$b_x = \sqrt{\omega_x/2}(x + ix/\omega_x), \tag{1}$$

and likewise for $y$. Then, we will have $b_x = f_x e^{-i\Omega t}/((\omega_x - \Omega) - i\Gamma/2)$ where $f_x \sim \cos(\varphi)$ is proportional to the force amplitude along $x$, for a linear drive along direction $\varphi$, and likewise for $b_y$ (Supplementary Notes 3 and 4 and Supplementary Fig. 4). Crucially, as we sweep the drive frequency $\Omega$, the phase lag between both linear polarizations (i.e., the phase of $b_y/b_x$) shifts. This leads to a transition from linear polarization to elliptical back to linear, even for a single pillar, as shown in Fig. 2a, b.

At first sight, it might seem surprising that elliptical motion patterns can emerge in this system, as they are not time-reversal invariant (selecting a sense of circulation), while both the bare model of an anisotropic oscillator and the linear drive itself conserve time-reversal symmetry. This is resolved by noting that

the phase lag leading to such motion only arises in the presence of dissipation, which does break time-reversal symmetry.

This theoretical description is borne out when observing a single pillar within the array (Fig. 2c–e). Both the spectrum (Fig. 2c) and the Poincaré sphere trajectory (Fig. 2d) show deviations from the idealized response of a single pillar, but this can be explained by the influence of the collective modes of the array. Apart from this, the overall features of the frequency evolution of the mechanical polarization (Fig. 2d, e) are consistent with two spectrally overlapping linear eigenmodes, showcasing the transition between the two orthogonal modes via an elliptical trajectory.

**Tight-binding model.** Based on our analysis of a single pillar and its polarization physics, we can now study the full array. Our theoretical analysis relies on a tight-binding model. In ref. [27], it has been shown experimentally that the coupling strength between pillars decreases with distance. Thus, in the model, we only consider the couplings between the nearest (side) and the next-to-nearest (diagonal) neighbors (Fig. 3a).

The interaction between neighboring pillars depends both on the relative vibration direction of the two pillars (Fig. 3b) and their distance. If the two pillars move perpendicular (parallel) to their line of connection, we call the interaction transversal (longitudinal), with coupling strength $J_{tt}$ ($J_{ll}$). Arbitrary anisotropies of any pillar can be fully characterized by introducing the frequencies $\omega_{x,y}$ and a coupling $J$ between $x$ and $y$ (cf. Fig. 3c).

In summary, the Hamiltonian of a $N \times N$ pillar array can be expressed in terms of the complex amplitudes $b_{x,y}$ (Eq. (1)) of the individual pillars as

$$H = \underbrace{\sum_{s,\mathbf{r}} \omega_{\mathbf{r},s} b_{\mathbf{r},s}^* b_{\mathbf{r},s} - J_{\mathbf{r}} b_{\mathbf{r},s}^* b_{\mathbf{r},\bar{s}}}_{\text{on-site Hamiltonian}} - \underbrace{J_{ll} \sum_{s,\langle\mathbf{r},\mathbf{r}'\rangle_s} b_{\mathbf{r},s}^* b_{\mathbf{r}',s}}_{\text{n.n longitudinal coupling}}$$
$$- \underbrace{J_{tt} \sum_{s,\langle\mathbf{r},\mathbf{r}'\rangle_{\bar{s}}} b_{\mathbf{r},s}^* b_{\mathbf{r}',s}}_{\text{n.n transversal coupling}} + \underbrace{H_d}_{\text{n.n.n coupling}}. \tag{2}$$

Here, $\mathbf{r} = (i,j)$ indicates the position of a pillar in the array, $s = \{x,y\}$ labels the direction of motion, and $\langle\mathbf{r},\mathbf{r}'\rangle_s$ indicates the nearest neighbors (n.n) in the $s$ direction. The bar symbol in the on-site Hamiltonian and the transversal coupling interchanges the two directions, i.e. $\bar{x} = y$ and vice versa. For a realistic

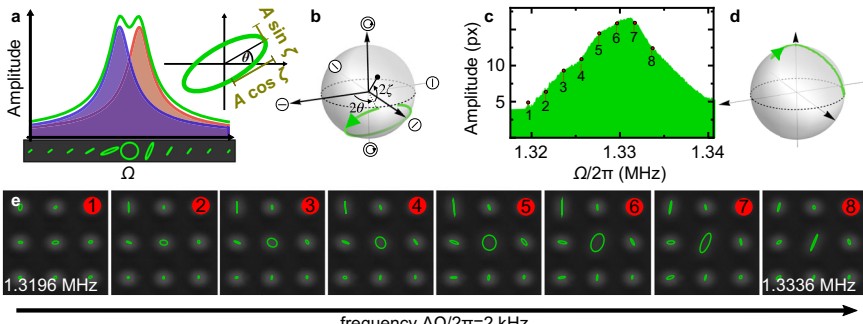

**Fig. 2 Polarization physics in a single nanopillar. a** Example of a theoretical frequency response for two separate orthogonal modes of a pillar (blue and red) and the combined response (green). The evolution of the trajectory along the frequency axis is indicated below the diagram. An additional inset shows the nomenclature for an arbitrary elliptical trajectory with semi-major (minor) axis length $A\cos\zeta$ ($A\sin\zeta$), and orientation of the major axis $\theta$. **b** Theoretical trajectory for the example in **a** on the Poincaré sphere. Note that the motion is (counter-)clockwise in the (upper) lower hemisphere. **c** Measured frequency response of a pillar with nearly degenerate modes. The amplitude is expressed in units of a camera pixel, where 1 px corresponds to approx. 28 nm. The corresponding path on the Poincaré sphere is shown in **d**. Note that we cannot measure the circulation sense of the ellipse in the experiment, therefore the path is depicted on the upper hemisphere for convenience. **e** Experimental evolution of the pillar's trajectory with drive frequency in the center of the images. The central pillar indicates the transition between the two orthogonal vibration directions via an elliptical trajectory. Frequency steps between neighboring images are $\Delta\Omega = 2\pi \times 2$ kHz.

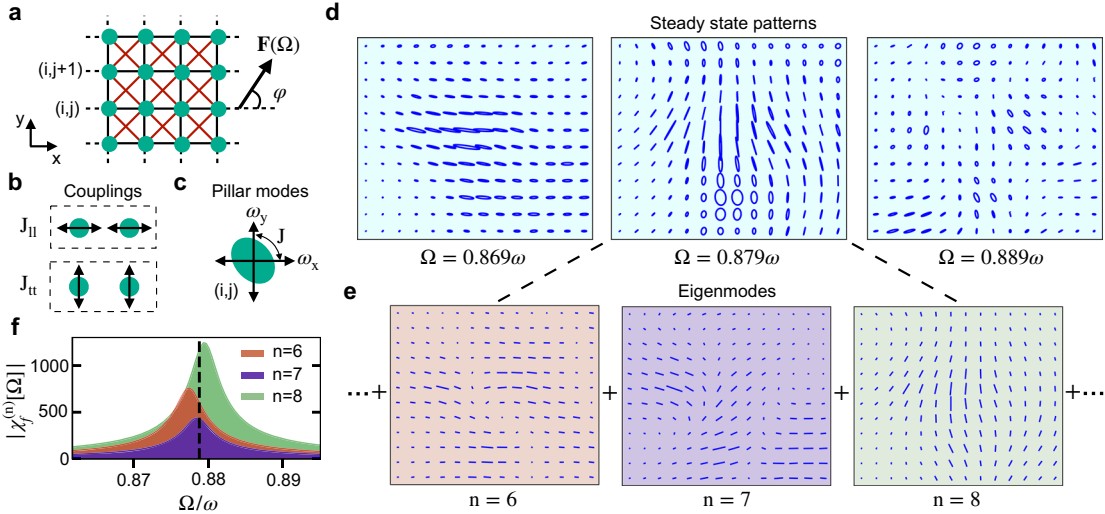

**Fig. 3 Tight-binding model for coupled pillar dynamics. a** Schematics of a pillar array with the nearest (black) and the next-to-nearest (red) neighbor couplings. **b** Sketch of the two types of couplings between the neighboring pillars. **c** Modeling an isolated pillar as two coupled harmonic oscillators with frequencies $\omega_{x,y}$ and coupling $J$. **d** Steady-state motion in a section of the array for various driving frequencies $\Omega$ renormalised with mean pillar frequency $\omega$. Each pattern is composed of a linear superposition of all the eigenmodes with their appropriate susceptibility $|\chi_f^{(n)}[\Omega]|$. **e** Three highest contributing eigenmodes for the steady-state pattern at $\Omega = 0.879\omega$. Notice that the eigenmodes contain only linear motion, thus the steady-state pillar trajectory can be elliptical only if more than one eigenmode contributes to it. **f** Contribution of each eigenmode in **e** to the steady-state pattern as a function of $\Omega$. The dashed line indicates $\Omega = 0.879\omega$. Parameter values of the tight-binding model are given in Supplementary Table 1.

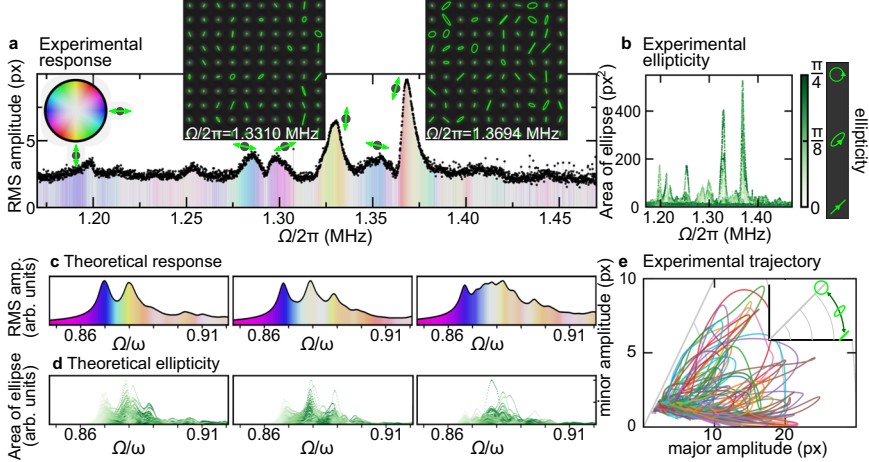

**Fig. 4 Evolution of polarization patterns. a** Measured RMS amplitude for the pillars in the array vs. drive frequency $\Omega/2\pi$. The color wheel illustrates the orientation $\theta$ of the pillar trajectories averaged over the whole array (more intense colors indicate larger homogeneity of the orientation). The small insets at each peak reflect the average orientation at the maximum of the peak. The experimental steady-state patterns at the two largest peaks are shown as further insets. **b** Experimental area of the elliptical trajectory for different drive frequencies with color-coded ellipticity $\zeta$. The pictograms next to the color bar illustrate how ellipticity corresponds to the actual form of the trajectory. **c** Theoretical RMS amplitude for different frequencies displayed in three different random realizations with the same disorder parameters. The color reflects the mean orientation of the array at each frequency according to the color wheel in **a**. **d** Theoretical area of the elliptical trajectory depending on the drive frequency with color-coded ellipticity as in **b** for the three disorder realizations in **c**. **e** Experimentally determined minor amplitude vs major amplitude of each pillar in the array around the largest peak: (1.36 MHz $\leq \Omega/2\pi \leq$ 1.38 MHz; data have been smoothed, see Supplementary Note 2 and Supplementary Fig. 3). As shown in the inset, trajectories become more elliptical when moving up from the x-axis to the diagonal. Tight-binding parameter values corresponding to **c** and **d** are given in Supplementary Table 1.

analysis of the experiment, this model is supplemented by a description of the disorder, as shown in the Supplementary Note 5 and Supplementary Fig. 5 (together with the explicit form of the next-to-nearest neighbor (n.n.n) coupling terms in Supplementary Note 6).

The steady-state response of the array can then be understood by decomposing into contributions from all the eigenmodes, cf. Fig. 3d–f (Methods).

**Evolution of polarization patterns.** With this theoretical model in hand, we can now study the experimentally observed frequency-dependent polarization patterns of the array, where we focus on the central 16 × 16 pillars, to avoid boundary effects (see Supplementary Note 7). This is in contrast to Fig. 2, where we were interested in the dynamics of a single pillar.

In Fig. 4a, we show the experimentally observed steady-state patterns and the RMS amplitude $\sqrt{\sum_{\mathbf{r}} A_{\mathbf{r}}^2}/N$ as a function of the

drive frequency $\Omega$. The amplitude response peaks at certain frequencies, as opposed to observing an uninterrupted band extending over all the eigenfrequencies of the array. This is because only eigenmodes with predominantly long-wavelength contributions couple constructively to the spatially uniform drive such that only the lower end of the frequency band and hence its first few modes are experimentally accessible. In other words, if we would have a spatially non-uniform drive, then we should see a much larger frequency band than what we observe. In addition, the two strongest peaks feature elliptical pillar motions (see insets of Fig. 4a), hence according to our earlier analysis there must exist at least two (linearly polarized) array eigenmodes within the bandwidth of these peaks. A well-established useful quantity in polarization physics is the complex Stokes field $\sigma = A^2 \cos(2\zeta)e^{2i\theta}$ (Fig. 2a, b). By studying its average across all pillars, $\sum_{\mathbf{r}} \sigma_{\mathbf{r}} / \sum_{\mathbf{r}} A_{\mathbf{r}}^2$, we can extract both the mean orientation (via the phase) and its fluctuations (via the magnitude). In Fig. 4a, these quantities have been color-coded to illustrate the evolution with frequency.

We now go beyond average quantities and study the distribution of individual ellipticities $\zeta$ across all the observed $16^2$ pillars. The resulting scatterplot (Fig. 4b) reveals that the majority of elliptical trajectories are observed at the two strongest resonances. It is equally illuminating to track the frequency evolution of attributes like minor and major axis of each pillar Fig. 4e, which clarifies that strongly elliptical motion is confined to a handful of pillars only.

It is not practicable to extract the (large) number of tight-binding model parameters from the experimental data, but fortunately many of our observations can still be qualitatively captured very well by the theoretical model. The effects of disorder are neatly illustrated by running numerical simulations on nominally identical parameters, but for different disorder realizations (Fig. 4c, d). On the one hand, this shows relatively significant fluctuations, but on the other hand, robust features can be identified. For instance, in agreement with the experimental observations, some of the resonances are primarily linearly polarized, while others support the elliptically polarized motion patterns discussed above.

All in all, the findings of Fig. 4 convincingly demonstrate the existence of collective motional polarization patterns in the nanopillar array. The observed vibrational patterns can not be explained by the independent co-vibration of individual pillars, but require the coupling between adjacent pillars of the array.

**Topological singularities in mechanical polarization patterns.** By virtue of setting up hundreds of coupled nanopillars in an array and the fast optical imaging measurement technique, our system opens the door to investigating the general area of complex spatial polarization patterns. This includes, in particular, the analysis of topologically robust polarization features[1,2]. The scenario that underlies the study of those mathematical concepts is the one realized both in the present work but also when observing electromagnetic waves projected on a planar cross-section: a monochromatic time-dependent vector field in two space dimensions. For such a situation, the polarization pattern can contain two elementary features. The first are isolated points of perfectly circular polarization, so-called 'C points', and the second are 'L lines', i.e. curves of linear polarization (Fig. 5a, b). When the polarization pattern is distorted smoothly, e.g. by changing the frequency of the waves or the geometry in which they propagate, both of these features are in general stable. This means the C points and L lines will move around but cannot disappear. The only exception is when L lines touch each other or contract to a point, or when C points meet. This robust behavior is understood

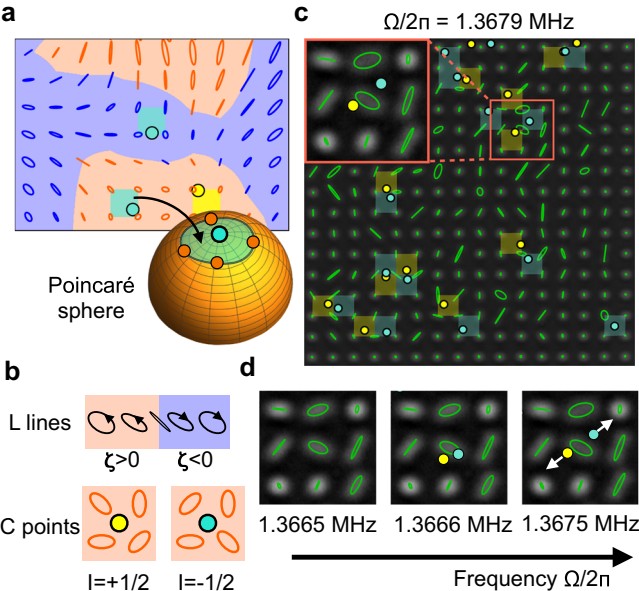

**Fig. 5 Topological singularities in mechanical polarization patterns.**
**a**, **b** Illustration of L lines and C points as topological singularities in a simulated motion pattern (with disorder level reduced compared to the experiment for easier visualization). A plaquette likely contains a C point if it maps to a patch on the Poincaré sphere that encloses the pole. **c** Experimentally observed pattern with algorithmically suggested candidates of C points (Supplementary Note 9 and Supplementary Figs. 8 and 9). Their topological winding index $I$ (winding of the orientation $\theta$ around the C point) is indicated by color. **d** A frequency sweep reveals a situation where a pair of C points of opposite index are created and move apart. Note that the algorithm achieves sub-lattice-spacing resolution via interpolation, and estimated trajectory data have been smoothed in **c** and **d**, (Supplementary Note 2 and Supplementary Fig. 3).

as a consequence of topology[1,2]. Polarization patterns of wave fields thus constitute one of the areas in physics where topology helps to gain insights. We emphasize that this is conceptually distinct from the investigation of topological band structures and transport along edge states that has drawn much attention recently (for a mathematical link between the two fields, see ref. [29]).

In our nanomechanical platform, we focus on the experimental observation of C points, since they can be extracted more reliably. One example is clearly visible in the experimental data of Fig. 2e, panel 5, where a particular nanopillar exhibits a clear circular trajectory.

Even when the distinction between left- and right-circular polarization is already accounted for, C points still can be distinguished by a further binary feature, the so-called winding index[2], somewhat similar to vortices in a phase field (Fig. 5b). We can make use of this to extract the locations of C points by investigating the winding of the ellipse orientation $\theta$ around any plaquette of the array. This technique is able to indicate the plaquettes of nonzero winding number, which are shown in Fig. 5c, for a particular drive frequency. As explained in the Supplementary Note 9 and Supplementary Fig. 10, these are topologically robust by mathematical necessity, i.e., they can only annihilate pairwise, which we also observe in the data. Going beyond that, we can even suggest the more detailed location of C points inside each plaquette, although this requires interpolating between the discrete grid points and thus depends to some degree on the interpolation scheme. We overcome the challenges posed by the disorder and measurement noise via careful data analysis,

including both averaging and numerical simulations confirming the robustness of the data analysis results (Supplementary Notes 2, 9 and Supplementary Figs. 3, 8, and 9).

By employing this automated procedure, we extract the evolution of C points from the experimental data, e.g. while sweeping the drive frequency. One example is shown in Fig. 5d, where we identify a pair-creation event of C points of opposite topological index $I = \pm 1/2$. We can also study systematic trends, such as the evolution of the number of C points with frequency (Supplementary Note 10 and Supplementary Fig. 11 for one example). In the future, stroboscopic imaging could reveal the handedness of the motion, which would allow reliable detection of L lines as well, since they separate areas of different handedness.

## Discussion

In summary, we have observed polarization patterns with signatures of collective dynamics in a two-dimensional array of coupled nanomechanical pillar resonators. Our measurements have been enabled by a whole-array optical imaging approach that allowed us to track the evolution of motional patterns with drive frequency. The platform introduced here enables the exploration of polarization fields in nanomechanics, unlocking phenomena for the domain of mechanics that in the past few years have led to a great number of insights and applications in electromagnetic systems. We have also discovered first indications of topological singularities in mechanical polarization fields.

In future experiments, the nanopillars can be even more strongly driven, which would permit the exploration of the collective motion in the nonlinear regime. In a different vein, the platform's flexibility in fabrication will naturally allow for the exploration of other actuation and measurement modalities. For example, the possible coupling of the array to electrical circuits or optical modes promises to both exploit alternative sensing techniques as well as optomechanical manipulation, up to and including the excitation of limit cycles composed of collective polarization patterns. Moving to a slightly different material platform, one could embed quantum dots inside the pillars, which could serve as light sources, where the mechanical drive could be used for beam steering applications. Highly parallelized vectorial force sensing suggests itself naturally as another potential application of the platform presented here.

In addition, one might implement more complex lattice geometries, such as variants of a honeycomb structure, which could be used to study entirely different effects, such as topological transport (e.g., in the valley Hall effect) and their interplay with collective polarization physics. Indeed as explained in[29], if our platform were modified to introduce time-reversal symmetry breaking (and reduced disorder), it would also allow the observation of band structure topology via the monitoring of real-space polarization patterns as a function of wave vector. In this context, individual actuation of pillars could be achieved for example by a photothermal drive. This would then allow to observe the propagation of wave packets through the array or along edge channels. In general, the investigation of time-dependent intrinsically non-periodic motion might be possible using a stroboscopic optical imaging scheme, if used in conjunction with periodically repeated reproducible excitations. As with any array platform there can of course be considerable disorder effects, but we managed to reduce these by careful control of the fabrication. Further improvements would enable more detailed studies of topological polarization phenomena. On the other hand, one might choose to focus on issues like Anderson localization, by deliberately increasing the disorder strength.

## Methods

**Fabrication details**. The conically inverted GaAs nanopillars (cf. Fig. 1a) are fabricated in a top-down fabrication process from a (100) GaAs wafer. The two-dimensional pattern of the array is defined via electron-beam lithography. This allows for a dense spatial integration of the pillars and a high control over the array geometry. A subsequent $SiCl_4/N_2$ anisotropic reactive-ion etch with protective etch mask of $SiO_2$ yields an array of high aspect ratio nanopillars.

**Imaging setup**. We measure the pillars' response to an external drive at room temperature and atmospheric pressure. The external periodic force is applied by a shear piezo glued underneath the sample. The response of the pillars to this drive is then imaged from above the sample (see Supplementary Note 1 and Supplementary Fig. 1 for more details). Resting pillars are identified as bright circles. Moving pillars appear smeared out compared to the resting pillar. The image of a moving pillar captures the envelope of its vibrational motion, as the exposure time of the camera, which is in the range of a second, greatly exceeds the oscillation period.

**Dynamical response**. All pillars in the array are subjected to an identical harmonic drive at frequency $\Omega$ and angle $\varphi$ with respect to the x-axis. Thus, the driving rate $f_x(f_y)$ is proportional to $\cos\varphi(\sin\varphi)$. For a mechanical damping $\Gamma$ (assumed identical for all pillars), the equation of motion of a pillar in the array is given by

$$\frac{db_{\mathbf{r},s}}{dt} = -i\frac{\partial H}{\partial b_{\mathbf{r},s}^*} - \frac{\Gamma}{2}b_{\mathbf{r},s} + if_s e^{-i\Omega t}. \tag{3}$$

The partial derivative $\partial H/\partial b_{\mathbf{r},s}^*$ is taken only over $b_{\mathbf{r},s}^*$, while $b_{\mathbf{r},s}$ is held as a constant. The steady-state solution can be written as a superposition of all the eigenmodes $b_{\mathbf{r},s}^{(n)}$ (eigenmode index labeled by n) of the Hamiltonian $H$ (cf. Fig. 3d, e) as

$$b_{\mathbf{r},s} = \sum_{n=1}^{2N^2} \chi_f^{(n)}[\Omega]b_{\mathbf{r},s}^{(n)}e^{-i\Omega t}, \chi_f^{(n)}[\Omega] = \frac{\sum_{s',\mathbf{r}'}b_{\mathbf{r}',s'}^{(n)*}f_{s'}}{(E_n - \Omega) - i\Gamma/2}. \tag{4}$$

Here, the mechanical susceptibility $\chi_f^{(n)}[\Omega]$ depends on both the drive frequency $\Omega$ and the overlap of the eigenmode with the drive (cf. Fig. 3f).

## Data availability

The data supporting the results presented in this article are available at Zenodo open-access repository under [https://doi.org/10.5281/zenodo.6127743][35].

## Code availability

The code supporting the results presented in this article are available at Zenodo open-access repository under [https://doi.org/10.5281/zenodo.6127743][35].

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

## Acknowledgements

The authors gratefully acknowledge technical support from H. Lorenz in the reactive-ion etching of the nanopillar arrays at LMU Munich. T.S. acknowledges support from the European Unions Horizon 2020 research and innovation program under the Marie Sklodowska-Curie grant agreement No. 722923 (OMT). F.M., J.D., and E.M.W. acknowledge support from the European Unions Horizon 2020 Research and Innovation program under Grant No. 732894, Future and Emerging Technologies (FET)-Proactive Hybrid Optomechanical Technologies (HOT). J.D. and E.M.W. acknowledge funding from the German Federal Ministry of Education and Research through contract no. 13N14777 funded within the European QuantERA cofund project QuaSeRT.

## Author contributions

J.D., F.M., and E.M.W. came up with the concept and planned the experiment. J.D. performed the pillar fabrication and measurements. P.P. developed the fabrication process and contributed to the implementation of the direct imaging setup. T.S. worked out most of the theory, with help from T.F. J.D., T.S., T.F., F.M. and E.M.W. analyzed the data and contributed to the writing of the manuscript. F.M. and E.W. supervised the project.

## Funding

## Competing interests

The authors declare no competing interests.
