## [Peer review file · Nature Communications]

REVIEWERS' COMMENTS

Reviewer #1 (Remarks to the Author):

J. Doster et al. have responded to some of my comments for their prior version of the manuscript. However, I am not satisfied with their answers to my questions. There indeed exists some gap between experiments and theory due to disorder, but the theoretical prediction in Fig. 3 here only discusses there may exist polarization and does not provide an experimental guidance for the rest of the paper. Indeed, I agree that the authors provide a different method for exploring collective behaviors of mechanical resonators. However, such a difference does not lead to the level of innovation usually required for publication in Nature Communications.

More specifically, below are my comments on their response for them to improve the manuscript for possible publication elsewhere:

On Point 2, 3, and 10: That is a full understatement. It is important to clarify what controllable and interesting phenomena of polarization can be demonstrated in coupled resonator arrays. As mentioned, the polarization pattern may simply be a linear superposition of eigenmodes of the 2D array if there is no controllable coupling between the vibrations along the orthogonal directions. Section 3, 4, and 5 of the Supplementary Materials elaborate that the 2D arrays can be treated as a scalar treatment within the current scheme and there are no vectorial features of polarization degree of freedom. In that case, the polarization degree of freedom in this array is not different from other degrees of freedom (e.g., supercell or vibration modes), leading to a weak practical significance. The authors should discuss its limitations and provide an outlook for future improvement.

On Point 4 and 6: It is true that optical imaging has its advantages for only collective motions. Nevertheless, the authors should also have an outlook for extensions of their approach to other potential applications, e.g., time-resolved observation.

Reviewer #2 (Remarks to the Author):

I have read the responses of the authors to my comments and those of the other referees, as well as the changes made to the manuscript. Overall, I believe their responses are clear and the changes made have

more clearly described the novelty and importance of the work, and specifically of studying these phenomena in the domain of nanomechanical arrays. I had in my previous report already motivated my appreciation of the originality and technique development. I thus believe this work is suitable for publication in Nature Communications.

I appreciate that they clarified the distinction of the physics that they study from topological insulators; that avoids potential confusion about the context and merits of the work. I also appreciate that they took my technical comment seriously and updated their algorithm. Also the physical significance of studying C points in this parameter regime was clarified by the addition of a new section in the SI.

As a remaining minor comment, I am less convinced by the answer of the authors to my question about a sentence in the abstract, where they state that "arrays investigated so far represent scalar fields on a lattice". It is true that in refs. 11 and 14 the collective states of orthogonal polarization decouple (in fact because of low disorder there), and that if one then investigates a single of those states the physics of polarization can be ignored. But the origin of those states is importantly linked to the fact that the arrays exhibit a polarization degree of freedom, i.e. degenerate modes with different polarization. For that reason, I believe the aforementioned statement in the abstract, as formulated now, is thus not accurately representing those works, and is also unnecessarily strict. I would recommend revising the wording. This is however a minor point, that does not affect my recommendation to publish this interesting work in Nature Communications.

Reviewer #3 (Remarks to the Author):

We think the authors have answered to the referees' comments as much as possible. It is likely that the complexity of the structure and the effect of disorder with non-unique behaviors do not allow for further conclusions at this stage. We would suggest that the authors highlight better ref. 29 (NJP (2017)) which contains the main theoretical basis of their discussion about topological polarization singularities, in particular in the Supplementary Information section.

Replies to the reviewer comments

Replies to reviewer #1

The reviewer wrote:

J. Doster et al. have responded to some of my comments for their prior version of the manuscript. However, I am not satisfied with their answers to my questions. There indeed exists some gap between experiments and theory due to disorder, but the theoretical prediction in Fig. 3 here only discusses there may exist polarization and does not provide an experimental guidance for the rest of the paper. Indeed, I agree that the authors provide a different method for exploring collective behaviors of mechanical resonators. However, such a difference does not lead to the level of innovation usually required for publication in Nature Communications.

More specifically, below are my comments on their response for them to improve the manuscript for possible publication elsewhere:

On Point 2, 3, and 10: That is a full understatement. It is important to clarify what controllable and interesting phenomena of polarization can be demonstrated in coupled resonator arrays. As mentioned, the polarization pattern may simply be a linear superposition of eigenmodes of the 2D array if there is no controllable coupling between the vibrations along the orthogonal directions. Section 3, 4, and 5 of the Supplementary Materials elaborate that the 2D arrays can be treated as a scalar treatment within the current scheme and there are no vectorial features of polarization degree of freedom. In that case, the polarization degree of freedom in this array is not different from other degrees of freedom (e.g., supercell or vibration modes), leading to a weak practical significance. The authors should discuss its limitations and provide an outlook for future improvement.

We thank the reviewer for reassessing our manuscript. The reviewer states that he/she is still not satisfied with our responses. We have considered the reviewer comments carefully. Respectfully, we think that there still may be a certain misunderstanding regarding the model used to describe the dynamics of the coupled nanopillar array, which we address in the following.

A scalar treatment of the vibrations along the x and y directions, or any other pair of decoupled orthogonal directions, is not possible in our system. Inspecting the model of Eq. (S-16) of SI section 5, one can observe the terms making such a treatment impossible, which are the contributions of the rotation matrices $R(a)$ and $R(y)$. Because the angles a and y vary between the lattice sites, it is not possible to decompose the system into two subsystems with motion in orthogonal directions. Polarization physics is thus essential in our model. This is in contrast to, for example, the effects of a hypothetical coupling term like $-J(b_x b_y + b_y b_x)$, which would merely correspond to a systematic position-independent anisotropy, where the motion in two eigendirections, $\square_2 (x + y)$ and $\square_2 (x - y)$, would be decoupled. To prevent our readers from having any such misunderstanding, we have now added a sentence in sections 3 and 5 of the SI.

On Point 4 and 6: It is true that optical imaging has its advantages for only collective motions. Nevertheless, the authors should also have an outlook for extensions of their approach to other potential applications, e.g., time-resolved observation.

We thank the reviewer for that comment. In the main text, we already had the following sentence to highlight the extension of our measurement approach: 'In the future, stroboscopic imaging could reveal the handedness of the motion, which would allow reliable detection of L lines as well, since they separate areas of different handedness.' We have now added the following additional observation about future possibilities, in the outlook at the end of the manuscript: 'In general, the investigation of time-dependent intrinsically non-periodic motion might be possible using a stroboscopic optical imaging scheme, if used in conjunction with periodically repeated reproducible excitations.'

Replies to reviewer #2

I have read the responses of the authors to my comments and those of the other referees, as well as the changes made to the manuscript. Overall, I believe their responses are clear and the changes made have more clearly described the novelty and importance of the work, and specifically of studying these phenomena in the domain of nanomechanical arrays. I had in my previous report already motivated my appreciation of the originality and technique development. I thus believe this work is suitable for publication in Nature Communications. I appreciate that they clarified the distinction of the physics that they study from topological insulators; that avoids potential confusion about the context and merits of the work. I also appreciate that they took my technical comment seriously and updated their algorithm. Also the physical significance of studying C points in this parameter regime was clarified by the addition of a new section in the SI.

We thank the reviewer for his/her positive assessment of our revised manuscript. We are pleased that the reviewer is satisfied with the proposed changes and clarifications.

As a remaining minor comment, I am less convinced by the answer of the authors to my question about a sentence in the abstract, where they state that "arrays investigated so far represent scalar fields on a lattice". It is true that in refs. 11 and 14 the collective states of orthogonal polarization decouple (in fact because of low disorder there), and that if one then investigates a single of those states the physics of polarization can be ignored. But the origin of those states is importantly linked to the fact that the arrays exhibit a polarization degree of freedom, i.e. degenerate modes with different polarization. For that reason, I believe the aforementioned statement in the abstract, as formulated now, is thus not accurately representing those works, and is also unnecessarily strict. I would recommend revising the wording. This is however a minor point, that does not affect my recommendation to publish this interesting work in Nature Communications.

We thank the reviewer for this suggestion. To be more cautious, we have now slightly modified the wording of the sentence, to clarify that we do not imply polarization degrees of freedom were necessarily completely absent, but only that an effective scalar treatment is possible: 'From a general point of view, the arrays investigated so far can be effectively treated as scalar fields on a lattice. Moving to a scenario where the vector character of the fields becomes important would unlock a whole host of conceptually interesting additional phenomena, including the physics of polarization patterns in wave fields and their associated topology.'

Replies to reviewer #3

We think the authors have answered to the referees' comments as much as possible. It is likely that the complexity of the structure and the effect of disorder with non-unique behaviors do not allow for further conclusions at this stage. We would suggest that the authors highlight better ref. 29 (NJP (2017)) which contains the main theoretical basis of their discussion about topological polarization singularities, in particular in the Supplementary Information section.

We thank the reviewer for reassessing our manuscript, and are pleased to see that there are no further objections. We have addressed the remaining comment raised by the reviewer, and now cite ref. 29 in the outlook of the main text in the following manner: 'Indeed as explained in [29], if our platform were modified to introduce time-reversal symmetry breaking (and reduced disorder), it would also allow the observation of band structure topology via the monitoring of real-space polarization patterns as a function of wave vector.' Moreover, we have now also cited ref. 29 in section 9 of the SI.